# Glimepiride-Loaded Nanoemulgel; Development, In Vitro Characterization, Ex Vivo Permeation and In Vivo Antidiabetic Evaluation

**DOI:** 10.3390/cells10092404

**Published:** 2021-09-13

**Authors:** Fizza Abdul Razzaq, Muhammad Asif, Sajid Asghar, Muhammad Shahid Iqbal, Ikram Ullah Khan, Salah-Ud-Din Khan, Muhammad Irfan, Haroon Khalid Syed, Ahmed Khames, Hira Mahmood, Asim Y. Ibrahim, Amani M. El Sisi

**Affiliations:** 1Department of Pharmaceutics, Faculty of Pharmaceutical Sciences, Government College University, Faisalabad 38000, Pakistan; fizzaabdulrazzaq33@gmail.com (F.A.R.); sajuhappa@gmail.com (S.A.); ikramglt@gmail.com (I.U.K.); manipharma1@gmail.com (M.I.); hiramehmood1993.hm@gmail.com (H.M.); 2Department of Pharmacology, Faculty of Pharmacy, The Islamia University of Bahawalpur, Bahawalpur 63100, Pakistan; asif_pharmacist45@yahoo.com; 3Department of Clinical Pharmacy, College of Pharmacy, Prince Sattam bin Abdulaziz University, Al-kharj 11942, Saudi Arabia; m.javed@psau.edu.sa; 4Department of Biochemistry, College of Medicine, Imam Mohammad ibn Saud Islamic University (IMSIU), Riyadh 11432, Saudi Arabia; sdikhan@imamu.edu.sa; 5Department of Pharmaceutics and Industrial Pharmacy, College of Pharmacy, Taif University, P.O. Box 11099, Taif 21944, Saudi Arabia; a.khamies@tu.edu.sa; 6Faculty of Pharmacy, Omdurman Islamic University, P.O. Box 382, Omdurman 14415, Sudan; asimyousif2005@yahoo.com; 7Department of Pharmaceutics and Industrial Pharmacy, Faculty of Pharmacy, Beni-Suef University, Beni-Suef 62521, Egypt; amany.elsese@pharm.bsu.edu.eg

**Keywords:** glimepiride, nanoemulsion, inclusion complexed nanoemulgels, clove oil, xanthan

## Abstract

Glimepiride (GMP), an oral hypoglycemic agent is extensively employed in the treatment of type 2 diabetes. Transdermal delivery of GMP has been widely investigated as a promising alternative to an oral approach but the delivery of GMP is hindered owing to its low solubility and permeation. The present study was designed to formulate topical nanoemulgel GMP system and previously reported solubility enhanced glimepiride (GMP/βCD/GEL-44/16) in combination with anti-diabetic oil to enhance the hypoglycemic effect. Nanoemulsions were developed using clove oil, Tween-80, and PEG-400 and were gelled using xanthan gum (3%, *w*/*w*) to achieve the final nanoemulgel formulations. All of the formulations were evaluated in terms of particle size, zeta potential, pH, conductivity, viscosity, and in vitro skin permeation studies. In vivo hypoglycemic activity of the optimized nanoemulgel formulations was evaluated using a streptozocin-induced diabetes model. It was found that a synergistic combination of GMP with clove oil improved the overall drug permeation across the skin membrane and the hypoglycemic activity of GMP. The results showed that GMP/βCD/GEL-44/16-loaded nanoemulgel enhanced the in vitro skin permeation and improved the hypoglycemic activity in comparison with pure and marketed GMP. It is suggested that topical nano emulsion-based GMP gel and GMP/βCD/GEL-44/16 could be an effective alternative for oral therapy in the treatment of diabetes.

## 1. Introduction

Over the past few decades, diabetes mellitus (DM) has started to appear as one of the major causative factors of deaths globally [1]. According to the International Diabetes Federation, it is extending at a very alarming rate, with a total estimated increase of 151 million (2000) to 463 million (2019) followed by a probability of increasing up to 700 million in near future [2]. Though several hypoglycemic drugs are widely used in diabetes management, a complete cure for diabetes still remains distant because of the numerous intrinsic deficiencies and side effects associated with these drugs [3,4]. The optimum drug concentration that is required cannot reach focal regions due to proteolytic degradation and chemical instability in a harsh pH environment. In addition, conventional dosage forms cannot be adjusted to tackle extensive fluctuations in glucose that lead to severe hypoglycemia [5,6]. Furthermore, other difficulties with drug therapy such as problems in drug absorption, short half-life, low solubility, low bioavailability, and adverse effects on other organs also impairs treatment [7].

Glimepiride is a third generation sulphonylurea per oral hypoglycemic agent [8]. It exerts its effect through two mechanisms by increasing the production of intracellular insulin from the (pancreatic) beta cells and by enhancing the sensitivity of intracellular (insulin) receptors to insulin action [9]. Currently, GMP is one of the most prescribed drugs for the treatment of diabetes, but its (oral) administration presents numerous deleterious side effects, low efficacy, and patient noncompliance [9,10]. The major problem associated with GMP is its poor aqueous solubility. To overcome this problem, many studies have been conducted to enhance the solubility of GMP by using a cyclodextrin inclusion complexation [11].

In our previously published report, we have attempted to devise binary and ternary GMP complexes by employing different hydrophilic polymers (βCD and Gelucire-44/14). The prepared complexes were also characterized using scanning electron microscopy (SEM), infrared (FTIR), powder X-ray diffraction (XRD), and differential scanning calorimeter (DSC). It was concluded that the solubility of GMP was significantly increased by both the GMP/βCD (1:4) and GMP/βCD/GEL-44/16 (1:4, 10% *w*/*w*) complexes, but the highest solubility was reported with the ternary complex of GMP/βCD/GEL-44/16 [12]. The GMP/βCD/GEL-44/16 complex (1:4, 10% *w*/*w*) was selected for the present study [12].

In the last several years, the topical application of numerous antidiabetic agents has been widely investigated as a promising alternative to an oral approach since it can attain the therapeutic drug concentration through the skin at a controlled and pre-determined rate [13], bypassing the first pass effect, reducing the frequency of dosing, and the treatment is able to be stopped through simple removal [14]. However, a safe and efficient permeation not only depends on the physicochemical character of a drug but also on the physical properties and chemical composition of the carrier [15,16].

Owing to the dense cellular structure and lipophilic character of stratum corneum, most of the drugs are not eligible to be delivered through the skin using traditional transdermal carriers. The application of nanocarriers for enhancing the transdermal delivery of drugs has emerged as a highly valuable alternative [17]. Several nanocarriers of GMP that have been investigated in the last ten years include liposomes, nanoethosomes, and self-nanoemulsifying particles, which have been designed for enhancing the transdermal bioavailability by encapsulation or solubilization, enhanced drug permeability, and attaining the controlled release [18,19]. Topical nanoemulsion formulations are well known for their ability to enhance skin permeation by temporarily disrupting the highly organized structure of the skin, and the potential of the topical nanoemulsions to deliver GMP was still left unexplored. Previous studies have demonstrated the use of self-nanoemulsifying particles to increase the effectiveness of GMP, but the drawback of these transdermal carriers is that they are only appropriate for smaller molecules that can easily penetrate stratum corneum (SC) [20].

Clove oil obtained from the flower buds of *syzygiumaromaticum* has recently received a lot of attention for its antidiabetic potential, especially in case of type 2 diabetes [21,22,23,24]. Clove oil contains eugenol and acetyleugenol, which can increase the fluidity of the lipids found in SC and can act as a penetration enhancer [24]. Advantage of using clove oil in order to deliver GMP is that is expected to improve the permeability and hypoglycemic activity with fewer side effects and higher efficacy. The antidiabetic activity of GMP using clove oil as an oil phase has not been investigated so far. Therefore, a nanoemulsion-based GMP gel using clove oil was formulated.

Hence, in order to enhance the ability of GMP to permeate the skin and to provide a more efficient therapeutic effect in comparison to previously reported studies, a nanoemulsion-based gel of glimepiride was designed. The unique feature of our study is that we introduce a new concept for the incorporation of solubility enhanced GMP (GMP/βCD/GEL-44/16) into the nanoemulsion-based gel (nanoemulgel) and evaluate its permeation and hypoglycemic activity against pure GMP and a marketed brand.

## 2. Materials and Methods

### 2.1. Materials

Glimepiride (GMP) was obtained as a gift from Saffron Chemical Industries Ltd. (Pakistan). Tween 20, Tween 40, Tween 80, Span 20, Span 80, propylene glycol, and PEG-400 were purchased from Daejung (Korea), whereas Labrafil-M 2125 CS, Capryol 90, and Gelucire 44/16 were gifted by Gatefosse (France). Ethanol, butanol, propanol, and glycerol were obtained from Merck (Pakistan). Clove oil, eucalyptus oil, peppermint oil, sandalwood oil, cinnamon oil, sesame oil, black seed oil, and rosemary oil were procured from Co Natural and Go Natural (Pakistan). The other chemicals and reagents used in this study were of analytical grade.

### 2.2. Solubility-Studies

The solubility of GMP was checked in numerous vehicles, which included antidiabetic oils, surfactants, and co-surfactants. For evaluating the solubility, two grams of these vehicles were taken in falcon tubes to which a surplus amount of glimepiride was integrated. These falcon tubes were then vortexed for ten minutes using a vortex mixer and were placed in shaking water bath for 48 h at 37 °C and 75 rpm. After 48 h, the falcon tubes were removed from the shaking water bath and were centrifuged at 3000 rpm for 15 min followed by filtration using a syringe-filter with a pore size of 0.45 µm. The obtained filtrate was then checked on with a UV-Visible spectrophotometer at 228 nm using methanol as blank. The same reagent was also employed for dilution if required [25].

### 2.3. Construction of Pseudo-Ternary Phase Diagram

To draw a phase diagram, the water titration method was used In different falcon tubes, oil was thoroughly mixed with surfactant/co-surfactant at different weight ratios of 0.5, 9.5, 1.0:9.0, 1.5:8.5, 2.0:8.0, 2.5:8.5, 3.0:7.0, 3.5:6.5, 4.0:6.0, 4.5:5.5, 5.0:5.0, 6.0:4.0, 7.0:3.0, 8.0:2.0, and 9.0:1.0. Water was then added drop wise into each tube (at 26 ± 2° followed by 2 min of vortex mixing and was allowed to equilibrate after light magnetic stirring for 2 h. After equilibrium establishment, the mixtures were observed for numerous phases, namely transparent, translucent, milky, turbid, gel, and phase separation. The clear emulsion with good flowability was declared as the nanoemulsion. Chemix (cxse 700, ver 4.00) was used to construct the ternary phase diagram [26].

### 2.4. Preparation of Nanoemulsion Formulations

Both blank and glimepiride/glimepirde inclusion complex-loaded nanoemulsions were fabricated through the spontaneous emulsification method [27]. Blank nanoemulsions were formulated by mixing oil and Smix at the optimum formulae that had been chosen from phase diagram on the basis of the nanoemulsion region. After achieving complete mixing, water was added drop-wise to the mixture and was vortexed until a transparent formulation was obtained.

### 2.5. Preparation of Drug-Loaded Nanoemulsion and Nanoemulsion Gel Formulations

For drug-loaded formulations, an accurately measured quantity of GMP and GMP/βCD/GEL-44/16 were added to the oil phase, which was subsequently followed by the addition of Smix and vortexing to facilitate complete solubilization. Water was then added gradually with proper mixing to attain a clear and homogenous nanoemulsion. The effect of drug loading in terms of the transparency of the NE systems was also studied, and 0.4% *w*/*w* drug was loaded in the final preparations. For the development of nanoemulsion based gels, the prepared NE formulations of both the GMP and the GMP/βCD/GEL-44/16 (GMP/βCD/GEL-44/16 were gelled using xanthan gum as a gel polymer. For this, an accurately measured amount of xanthan gum (3 g) was dissolved in 100 mL of distilled water in order to formulate a gel base to which subsequent addition of triethanolamine was made to neutralize the pH. The drug and inclusion complex containing nanoemulsions were then added to this gel base in a ratio of 1:1 by magnetic mixing for a period of 15 min at a 250 rpm speed to properly acquire homogenized nanoemulsion based gels [28].

### 2.6. Characterization of Nanoemulsion and Nanoemulgel Formulations

For determining the particle size and polydispersity index of the prepared formulations, Malvern Zetasizer (ZS90, UK) with DLS (dynamic scattering technique) was used. All estimations were conducted at 25 °C and at an angle of 90°. Zeta potential (ELS, electrophoretic scattering) was determined using the same instrument at 633 nm by a 1 volt electric field application. All nanoemulsion formulations were diluted 100 times (distilled water) prior to particle size and zeta potential measurement, and separate samples were used for each evaluation. The viscosity of the prepared formulations was tested with a Brookfield viscometer (DVII+ Pro). The spindle used for checking the viscosity was s 63. It was rotated at a speed of 100 rpm for 1 min, and each sample was tested in triplicate. The pH of the both blank and drug/inclusion complex-loaded nanosized emulsion was checked with a pH meter (HI 9811-5 Hanna, United States). Electrical conductivity of both the blank and drug/inclusion complex-loaded formulations were also investigated, and the instrument used for this purpose was a portable EC meter (Hanna^®^ instruments HI 9811-5 Romania). Nanoemulgel formulations were characterized by tests similar to those performed for NE (pH and viscosity). The spreadability of each nanoemulgel formulation was also assessed using a glass plate according to the method reported by Dantus [29]. An accurately weighed amount of nanoemulgel was spread on a glass plate, which was pre-marked with a circle measuring 2 cm. The presenting assembly was then topped with another glass plate followed by a weight of 0.5 kg for five minutes. Each formulation was analyzed in triplicate (*n* = 3).

### 2.7. Ex Vivo Permeation Study

Animal study was conducted according to the guidelines of Animal Welfare Act and the those from the Care and Use of Laboratory Animals of National Institutes of Health. All of the protocols were thoroughly studied and consented to by the local Institutional Review Board of GCUF University (GCUF/ERC/2185 Dated: 23 April 2020) after evaluating and ensuring the proper use of animals. In order to conducted permeation studies, rats weighing in the range of 200–300 g were sacrificed, and the hairs on their abdominal area were removed followed by surgical removal of the skin. The excised skin was subsequently rinsed with alcohol and distilled water, and it was stocked at −20 °C until additional application. Before use, it was removed from the freezer and was thawed to room temperature followed by placement in phosphate buffer for at least 60 min in order to properly hydrate the membrane. For the permeation study, a Franz diffusion cell apparatus with two compartments named as donor and receiver compartment was used. As far as the skin is concerned, it was positioned in between the two compartments in such a way that the stratum corneum (SC) side countered the receiver compartment. The GMP dispersed in clove oil and a GMP aqueous suspension with (0.5%) carboxymethyl cellulose were used as controls. The formulations (1 gm, control and sample) to be tested were placed in donor compartment, whereas receiver compartment contained the dissolution medium, i.e., the phosphate buffer (pH 7.4) plus methanol (70:30, % *v*/*v*), which was constantly stirred with magnetic bars at 300 rpm. The temperature was regulated at 37 °C, and samples were taken at specified time points preceded by replacement with the same amount of dissolution medium. The samples were then scanned using a UV spectrophotometer at a wavelength of 228 nm. Experimentation was done in triplicate, and the obtained data were articulated as the cumulative quantity of GMP permeated across SC against time [30]. Furthermore, estimations of various skin permeation parameters, namely drug flux (Jss), as well as the permeability co-efficient (K_p_) were also determined.

### 2.8. In Vivo Antidiabetic Study

In order to inspect the anti-diabetic potential of the selected test formulations, male wistar rats weighing 200–250 g were used. All of the animals were accommodated under managed temperature conditions (24 °C) and humidity (70–30%) along with a proper diet consisting of typical rat feed and water. Diabetes mellitus was developed by injecting streptozocin intra-peritoneal at a dose of 50 mg/kg body weight. Prior to the diabetes generation, the rats were fasted for the night; however, only food was restricted, and water was available at all times. After 14 days, the serum blood glucose levels of fasted rats was checked using a glucometer, and the rats with glucose levels between 250–300 mg/100 dL were selected for further studies. The overall experimental design of this study involved six groups with each group consisting of five rats in total. The rats in Group I (normal control) and Group 2 did not receive any treatment at all and served the as normal and diabetic controls, whereas the rats in Group 3 (positive control) were given a per oral glimepiride dose of 10 mg/kg bodyweight [31]. Rats in Group 4 were given a blank gel formulation, which was prepared by incorporating blank nanoemulsion in xanthan gel. Similarly, the rats in Group 5 and 6 were subjected to the topical application of nanoemulgels, namely GMP loaded nanoemulgels and GMP/βCD/GEL-44/16-loaded nanoemulgels at a dose of 10 mg/kg bodyweight, respectively. For oral administration, the standard oral gavage technique was used. Blood samples were collected from the tip of the tail by nicking the lateral tail vein.

### 2.9. Skin Irritation Study

The skin irritation study was conducted according to OECD guidelines and for this purpose, the wistar rats weighing in the range of 150–200 g were used. All of the rats were divided completely at random in to four groups: Group 1 (no application), Group 2 (blank nanoemulgel, NEG), Group 3 (GMP-loaded NEG), and Group 4 (GMP/βCD/GEL-44/16-loaded NEG). The hairs on the dorsal surface of each rat were carefully removed by shaving the rats 24 h prior to experimentation, and 0.5 gm of each test formulation was applied onto the shaved skin for 4 h by evenly spreading it over an area of 6 cm^2^. The testing area was ascertained for any erythemic or edema reaction after 1, 24, 48, and 72 h of application. The experiential tolerance reaction was scored as 0, 1, 2, and 3 for no, slight, moderate, and severe erythema/edema, in that order. After 72 h, all of the animals were sacrificed through cervical decapitation, and the skin from the testing area was obtained. The obtained skins were stored in 10% neutral buffered formalin solution, fixed in paraffin, cut in 5 µm sections, and stained with H &E dye followed by imaging under light microscopy [32].

### 2.10. Statistical Analysis

All of the parameters were taken in triplicate, and the obtained results were presented as mean ± standard deviation. For statistical significance, one-way ANOVA, two-way ANOVA and Tukey’s multiple comparison test were employed in order to assess any significant differences amongst the quantitative variables. Data values having *p* < 0.05 were marked as statistically significant.

## 3. Results and Discussion

### 3.1. Solubility Studies

The solubility of GMP was evaluated in variety of antidiabetic oils, surfactants, and co-surfactants for the screening and selection of all of the nanoemulsion (NE) components. Solubility results are presented in Figure 1. The selection of an oil phase is highly critical, as it is accountable for loading the drug and maintaining it in a dissolved state in the nanoemulsion system indirectly [33]. Amongst all of the oils that were evaluated, the highest solubility was found to be observed with clove oil (22.96 ± 0.30 mg/mL); thus, it was selected as the oil phase. Surfactants and co-surfactants assist in reducing the interfacial tension between the two phases by adsorbing at their interface and provide a mechanical barrier against coalescence. In the case of surfactants, a non-ionic type is mostly preferred in comparison to their ionic counterparts that are often associated with numerous toxicity and bioactivity related concerns. Non-ionic surfactants are considered to be safe due to their hydrophilic nature [34]. Therefore, in this study, only nonionic surfactants were studied. In these surfactants, Tween-80 (19.2 ± 0.18 mg/mL) exhibited the uppermost potential in terms of solubilizing glimepiride and was therefore selected for further investigation. Besides solubility, Tween-80 also has an HLB value of 14.5, which ensured the corresponding O/W emulsion type [35]. Among the co-surfactants, PEG-400 (1.77 ± 0.15 mg/mL) was selected.

### 3.2. Pseudo Ternary Phase Diagram

A pseudo-ternary phase diagram was developed to identify the NE region and to select the optimum concentration of clove oil, Tween-80, and PEG-400 for the development of the nanoemulsion. The presented pseudo ternary phase diagram was constructed in the absence of GMP, and the S/Co-S ratio of 2:1 was employed on the basis of the solubility results. The NE region that was obtained is exhibited by the shaded area (Figure 2).

In nanoemulsion systems, the efficiency of the nanoemulsification could be estimated by the zone exhibited by its nanoemulsion region [36]. The larger the zone, the higher the efficiency will be. The pseudo-ternary phase diagram developed using water, clove oil, and Tween-80 and PEG-400 (2:1) exhibited a broad nanoemulsion region, which was found to increase with the increase of the quantity of the of the surfactant mixture. Furthermore, this increment was towards the clove oil and water axis, implying that the potential of water inclusion is strengthened as the ratio of surfactant mixture to clove oil elevates, subsequently leading to application in the NE area.

### 3.3. Selection of Formulae

Although unlimited formulations can be selected from the obtained NE region, only three formulas were selected for the development of nanoemulsion, which are presented in Table 1. It is well documented that the composition of a nanoemulsion (NE) can impact various attributes of a nanoemulsion; therefore, different concentrations of oil and surfactant mixtures were selected at 5% intervals. As reported in the literature, an increase in oil concentration may increase the droplet diameter of a NE, and the selection of oil in low concentrations was considered suitable (10%, 15%). Since surfactants, especially ionic surfactants, can induce irritation, non-ionic surfactants in lower concentrations were preferred. Thus, for every percentage of oil selected, the minimum concentration of Smix was adopted.

### 3.4. Characterization of Nanoemulsion and Nanoemulsion Based Gels

#### 3.4.1. Visual Inspection

All formulations (blank, GMP and GMP/βCD/GEL-44/16-loaded) showed complete transparency, and the addition of GMP and GMP/βCD/GEL-44/16 did not influence the color or stability of the system, which could be due to the presence of non-ionic surfactants, which are known for their ability to not be influenced by the pH or ionic strength of the integrated agent [37].

#### 3.4.2. Droplet Size and Polydispersity Index Measurement (PDI)

The results of the particle size and PDI determination are mentioned in Table 2. The mean particle size of all of blank formulations varied from 133.4–157.9 nm. In the case of the GMP and GMP/βCD/GEL-44/16-loaded preparations, particle sizes were between 141.9–213.3 nm and 143.1–233.3 nm, respectively.

It was observed that the particle size in formulation NE-1 and NE-2 increased as the content of clove oil increased from 10% to 15% *w*/*w*. An increase in the particle size with the increase in clove oil content was seen due to globule expansion in droplets [38]. While keeping the oil concentration constant, when the amount of surfactant and co-surfactant increased in formulation NE-3 from 40–45% *w*/*w*, the particle size decreased. The decrease in particle size could be attributed to the extortionate proportion of surfactant in opposition to the co-surfactant, as the surfactant and co-surfactant are involved in the condensation and expansion of the interfacial film, respectively [39]. The surfactant used in this system is Tween 80, which can also achieve small mean particle size, as it is an ester of LCFAs (long chain fatty acids) [40]. The appropriate amount of surfactant has an influence on the particle size.

The mean particle size for all of the prepared formulations (blank, GMP, GMP/βCD/GEL-44/16-loaded) was in the acceptable size range of 10–500 nm [41]. Several studies also reported that nanoemulsion formulations with a particle size below 500 nm are also suitable [42]. Thus, all formulations were considered to fall into the acceptable nanosize range.

The polydispersity index (PDI) determines the measure of particle size homogeneity. Normal values for PDI range from 0.0–1.0. The PDI values obtained for all of the formulations were less than 0.5 (Table 2), showing narrow and uniform size distribution [43].

#### 3.4.3. Measurement of Zeta Potential

Zeta potential is usually an indicator of the stability of a NE system. It is evident that higher zeta potential values ensure higher stability. However, these values are not reliable for the assessment of stability, as the literature presents a wide range of these values ranging from 1.5–45.5 mV [44,45]. The zeta potential values that were obtained for all of the NE formulations are presented in Table 3. Values for zeta potential were found to be negative, in the range of −12.8 to −20.7 mV. Nanoemulsions with a negative charge on their droplets experienced electrostatic repulsion, which guarantees free coalescence and a well-separated emulsion system [46].

#### 3.4.4. Determination of Viscosity

Viscosity values obtained for all of the nano formulations are presented in (Table 4). It was apparent from the data that the developed formulation had low viscosity for an O/W nanoemulsion system exhibiting a Newtonian flow pattern [47].

Table 4 exhibits an increase in the viscosity values when the concentration of clove oil (10–15%) and Smix increased (35–40%) and the water content decreased from 55 to 45% in the NE-1 and NE-2 formulations. The increase in viscosity was owed to a higher internal phase ratio in an emulsion system [48,49]. The viscosity of formulation NE-3 further increased when the amount of Smix increased and water concentration decreased. This increase could be credited to the highly hydrated hydrophilic chains of the surfactant Tween 80 that are strongly connected together through hydrogen bonding, allowing for stronger interaction [49,50].

#### 3.4.5. pH and Conductivity Measurements

pH and conductivity values of blank, GMP, and GMP/βCD/GEL-44/16-loaded formulations are presented in Table 5. The pH values obtained with the blank formulations ranged from 5.19 to 5.59, which exhibited a slight decrease from 5.10 to 5.51 and 5.06 to 5.45 after the incorporation of GMP and GMP/βCD/GEL-44/16. The pH of a normal adult skin falls between 4 to 6.5, but pH values ranging from 5 to 8 are unobjectionable for skin application [51]. A decrease in the pH values after the incorporation of drug ensured its acidic nature. An electrical conductivity test was used to find the type of emulsion. Typically, O/W nanoemulsion systems have higher electrical conductivity values (10–100 µS/cm) in compared to O/W nanoemulsion systems (10 µS/cm), owing to the exceptional conductivity properties of their outer aqueous phase [52]. Electrical conductivity values before GMP loading were between 63.38 to 35.72 µS/cm. After GMP and GMP/βCD/GEL-44/16 loading, these values ranged from 62.50 to 34.01 µS/cm and 60.7 to 34.19 µS/cm, respectively. The electrical conductivity values were found to decrease with the decrease in the amount of water in the system.

Spreadability, pH, and viscosity values for all of the nanoemulgel formulations are illustrated in Table 6. All of the formulations exhibited excellent spread behavior by applying small volume of shear, implying easy practical application from the patient point of view. The pH and viscosity values of these formulations were suitable for skin application.

#### 3.4.6. Ex Vivo Permeation Study

Figure 3 exhibits the in vitro permeation profile of both the GMP and the GMP/βCD/GEL-44/16-loaded nanoemulgels. All of the nanoemulgel formulations displayed a steady rise in GMP permeation over time. The results displayed that the GMP-loaded nanoformulations presented superior permeation when compared to GMP–clove oil and the GMP aqueous suspension. The overall permeation order that was observed was NEG(IC) > NEG> GMP–clove oil > GMP aqueous suspension. Among the GMP-loaded formulations, NEG-1 exhibited superior release than other formulations, which could be explained on the base of their comparatively smaller size and lower viscosity.

Similar patterns were seen with the GMP/βCD/GEL-44/16-loaded formulations. The permeation parameter (flux and permeability constant) values were greater than the GMP-loaded formulations. The rapid onset release may be due to the incorporation into a nano-carrier and close cutaneous contact [33]. The values of the permeation parameters that were obtained for all of the nanoemulgel formulations are presented in Table 7.

In vitro permeation is essential to predict the overall in vivo effect, as it not only involves the release of drug from the carrier but also its absorption into the skin. The difference in permeation profile could be due to a number of reasons. The highest permeability of NEG-1 for both GMP and GMP/βCD/GEL-44/16 was better attributed to its lowest size. Smaller droplet diameters offer a greater interfacial area (for dissolution) and establish closer dermal contact, allowing for higher drug penetration and thus elevated drug content at the target site [53]. Variation in the composition of the formulations could also govern the permeation. This is also backed by various studies, which reported that due to their higher aqueous and lower surfactant concentration O/W emulsions exhibit higher drug fluxes in comparison to W/O emulsions [54,55]. This behavior could be acknowledged as the thermodynamic activity of lipophilic drugs, which becomes considerable at lower surfactant concentrations [56]. Additionally, higher water concentration is also associated with better skin hydration, which causes the corneous cells to engorge, making the channels implicated in drug transport to expand. The hydration also causes disruption in the chains affixed to these corneous cells, consequently leading to disruption in the lipid bilayer and enhanced permeation [57].

The NE-1 of both the GMP and GMP/βCD/GEL-44/16-loaded gels exhibited the lowest viscosity because higher viscosity could retard the release of drug. This means that a permeation process is most likely to be dependent upon the composition of a nanoemulgel formulation. However, permeation is independent of any concentration gradient, as all of the nanoemulgel formulations carry an equivalent drug load. Nanoemulgel formulations consisting of GMP/βCD/GEL-44/16 expressed high flux values in comparison to GMP-loaded nanoemulgels. The high flux values could be due to the ability of the cyclodextrins to liberate more drug molecules from the complex. As a result, a higher amount of free drug is maintained, leading towards higher flux values [58]. Both drug flux and permeability constant values for all of nanoemulgel formulations were found to be highly significant (*p* < 0.05) when compared to the control groups (GMP in clove oil, GMP/βCD/GEL-44/16 in clove oil, and GMP aqueous suspension). This further validated enhanced permeation from the nanoemulgel formulations. Statistical evaluation of permeation data also revealed the significantly enhanced permeation parameters (flux and Kp) for NEG-1(IC) when compared to nanoemulgels without IC. However, other formulations did not show significant difference in these parameters for IC.

#### 3.4.7. In Vivo Anti-Diabetic Studies

In vivo hypoglycemic effects of the optimized nanoemulgel formulations were examined using streptozocin-induced diabetic rats. Appendix A and Figure 4 present the results of the anti-diabetic study which explains the multiple comparisons of the obtained blood glucose values at each time interval to its corresponding control at 0 h. Group 1 (normal control) and Group 2 (diabetic control) exhibited no noteworthy reductions at all when compared to their corresponding control group at 0 h and were found to be non-significant (*p* > 0.05). Group 3 (oral standard group) significantly (*p* < 0.05) reduced hyperglycemic levels from the initiation of therapy, i.e., after 2 h upon comparison to its relevant control at 0 h, but managed only managed to sustain it for 6 h. Groups 4, 5, and 6 displayed significant (*p* < 0.05) lowering in the blood glucose levels from 4 h, but Group 6 (GMP/βCD/GEL-44/16 NEG) showed the highest hypoglycemic effect in comparison. It is evident from the findings that the simultaneous delivery of clove oil and GMP from nanoformulations (Group 5, 6) exhibited higher hypoglycemic activity in comparison to blank nanoemulgels using clove oil only (Group 4). This enhanced hypoglycemic effect could be due to the improved solubility and in vitro permeability of the drug. However, a higher degree of retention was observed with the GMP/βCD/GEL-44/16 nanoemulgels that expanded the release period (up to 24 h) and changed the overall hypoglycemic effect. Furthermore, the role of beta-cyclodextrin as a permeation enhancer may also be considered [59]. Our findings were in agreement with previously reported studies that showed higher therapeutic activity with βCD [60]. As a result, NEG-1(IC) illustrated the greatest permeation outcomes.

#### 3.4.8. Skin Irritation Studies

The irritation scores obtained for all of the tested formulations were found to be null, denoting that there was no noticeable erythma or edema either on or around the site of application with both the blank and drug-loaded nanoemulgel preparations. It is presumed that the pH of a nanoemulgel is comparable to skin, as it produced no irritation reaction. Secondly, the inclusion of highly safe ingredients and gel base also assured safe topical application [17]. Consequently, these observations were also supported by the histopathological findings, as no markings of any irritation or inflammation were readily apparent in skin microscopy with either blank or drug-loaded nanoemulgels in comparison to control group, which demonstrated intact stratum corneum, collagen fibers, and appendages with no marking of any inflammatory cells, as visible in Figure 5.

GMP/βCD/GEL-44/16-loaded nanoemulgels effectively enhanced the permeation and ultimately the hypoglycemic activity of glimepiride without any negative effect on the skin. Furthermore, relying upon the higher antidiabetic activity of GMP/βCD/GEL-44/16 nanoemulgels, a possible reduction in GMP dose can also be recommended in the future.

## 4. Conclusions

Nanoemulgel systems consisting of clove oil as the oil phase, Tween 80 as a surfactant, and PEG-400 as a co-surfactant were successfully formulated and characterized. The preparation of nano formulations using clove oil to address the issues of poor transdermal bioavailability for compounds such as GMP have not been investigated before. The incorporation of solubility enhanced GMP/βCD/GEL-44/16 significantly improved the in vitro release and skin absorption of the glimepiride. GMP/βCD/GEL-44/16-based nanoemulgels also presented higher antidiabetic activity in comparison to pure GMP-based nanoemulgel and marketed GMP. These finding clearly suggest that enhanced solubility leads to an increase in bioavailability. It can also be concluded that a synergistic combination of nanoemulsion and gel provides an efficient and successful carrier for the topical delivery of both GMP and solubility enhanced GMP in rats.

## Figures and Tables

**Figure 1 cells-10-02404-f001:**
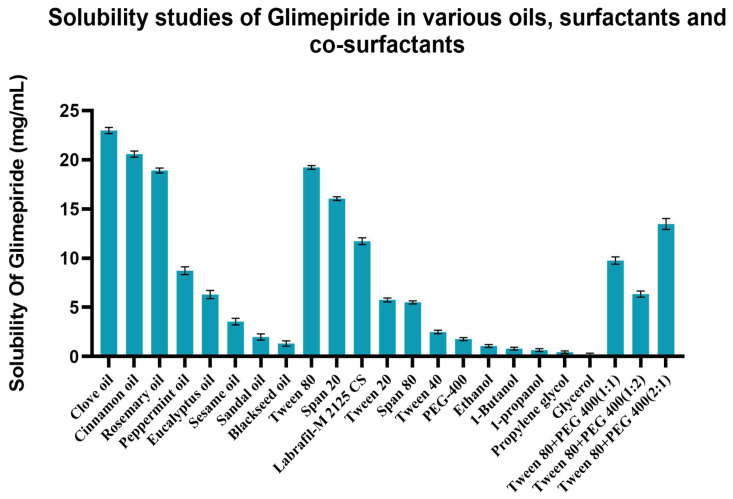
Solubility studies of glimepiride in various oils, surfactants, and co-surfactants (*n* = 3).

**Figure 2 cells-10-02404-f002:**
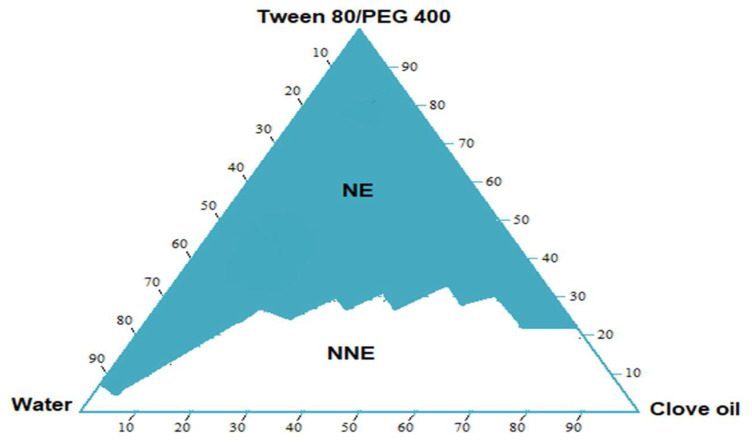
Pseudo-ternary phase diagram consisting of water, clove oil, and Tween 80 + PEG-400 (2:1).

**Figure 3 cells-10-02404-f003:**
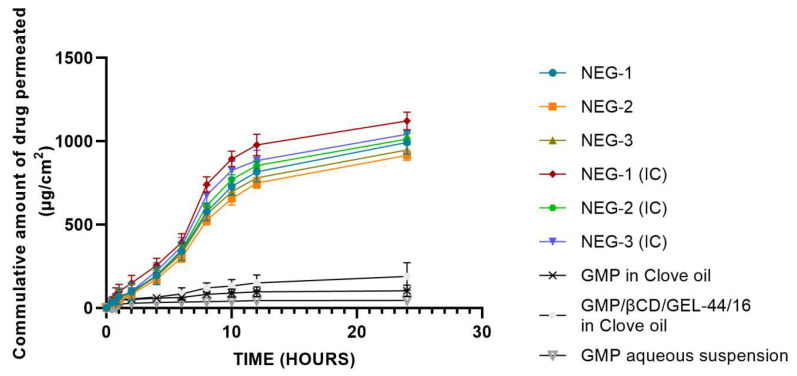
Permeation profile of GMP- and GMP/βCD/GEL-44/1-loaded nanoemulgels in contrast to their oily drug suspension.

**Figure 4 cells-10-02404-f004:**
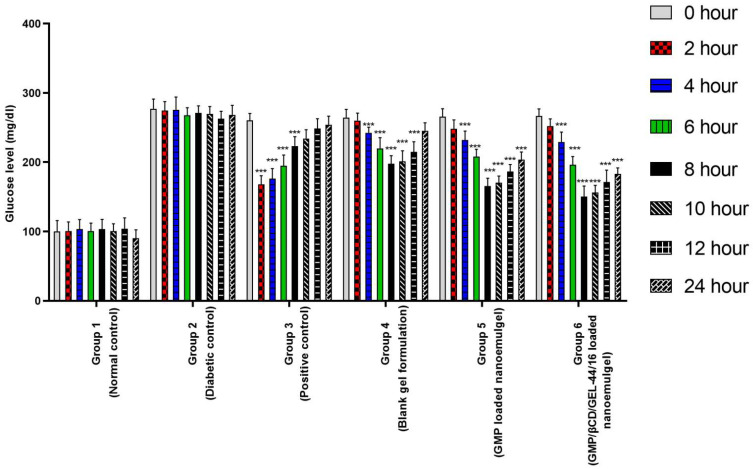
Hypoglycemic effect of GMP and GMP/βCD/GEL-44/16-loaded nanoemulgels and marketed oral glimepiride in streptozocin-induced diabetes model. (*** indicates *p* < 0.001).

**Figure 5 cells-10-02404-f005:**
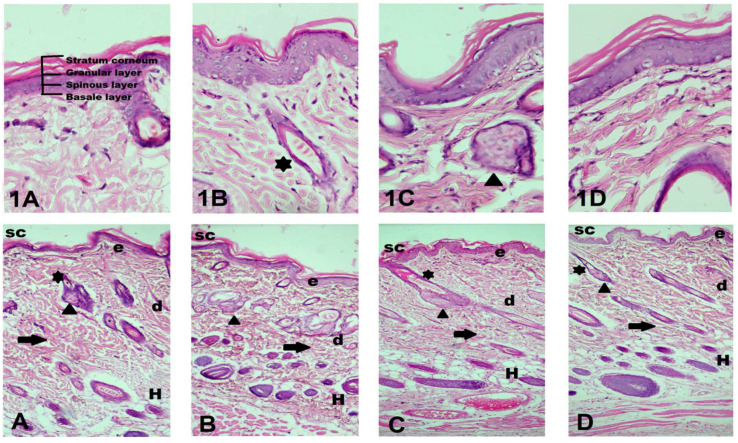
Histopathological examination of rat skin at 10× and 40× magnification. (**A**,**1A**) Group 1 no application; (**B**,**1B**) Group 2 Blank NEG; (**C**,**1C**) Group 3 NEG-1; (**D**,**1D**) Group 4 NEG-1 (IC). Skin structures: (SC) stratum corneum; (d) dermis; (H) hypodermis; (*) hair follicle; (arrow head) sebaceous gland; arrows indicate collagen fibers.

**Table 1 cells-10-02404-t001:** Optimum formulae for the preparation of nanoemulsion.

Formulation Code	Composition (% *w*/*w*)		
	Oil	Smix	Water
NE-1	10%	35%	55%
NE-2	15%	40%	45%
NE-3	15%	45%	35%

**Table 2 cells-10-02404-t002:** Particle size, polydispersity index (PDI) of blank (B), glimepiride-loaded (GMP), and inclusion complex-loaded (IC) nanoemulsions (Mean ± SD, *n* = 3).

Formulation Code	Particle Size (nm)			Polydispersity Index (PDI)		
	Blank	GMP-Loaded	GMP/βCD/GEL-44/16-Loaded	Blank	GMP-Loaded	GMP/βCD/GEL-44/16-Loaded
NE-1	133.4 ± 2.01	141.9 ±3.35	143.1 ± 2.60	0.269 ± 0.006	0.260± 0.004	0.315± 0.008
NE-2	157.9 ± 2.40	213.3 ± 2.55	233.3 ± 3.60	0.260 ± 0.006	0.428± 0.006	0.421± 0.007
NE-3	137.8 ± 2.60	151.9 ± 3.20	168.4 ± 2.20	0.230 ± 0.004	0.225± 0.008	0.383± 0.008

**Table 3 cells-10-02404-t003:** Zeta potential analysis of blank (B), glimepiride-loaded (GMP), and inclusion complex loaded- (IC) nanoemulsions (Mean ± SD, *n* = 3).

Formulation Code	Zeta Potential (mV)		
	Blank	GMP-Loaded	GMP/βCD/GEL-44/16-Loaded
NE-1	−17.5 ± 0.4	−20.5 ± 0.2	−14.2 ± 0.45
NE-2	−17.8 ± 0.3	−20.7 ± 0.35	−17.1 ± 0.35
NE-3	−16.5 ± 0.4	−15.4 ± 0.30	−12.8 ± 0.25

**Table 4 cells-10-02404-t004:** Viscosity analysis of blank (B), glimepiride-loaded (GMP), and inclusion complex-loaded (IC) nanoemulsions (Mean ± SD, *n* = 3).

Formulation Code	Viscosity (cP)		
	Blank	GMP-Loaded	GMP/ΒCD/GEL-44/16-Loaded
NE-1	30.4 ± 1.85	32.8 ± 1.80	34.7 ± 1.53
NE-2	43.7 ± 1.50	45.2 ± 1.51	46.4 ± 1.12
NE-3	55.1 ± 2.20	58.1 ± 1.60	58.7 ± 1.15

**Table 5 cells-10-02404-t005:** pH and electrical conductivity investigation of blank (B), glimepiride-loaded (GMP), and inclusion complex-loaded (IC) nanoemulsions (Mean ± SD, *n* = 3).

Formulation Code	pH			Conductivity (µS/cm)		
	Blank	GMP-Loaded	GMP/βCD/GEL-44/16-Loaded	Blank	GMP-Loaded	GMP/βCD/GEL-44/16-Loaded
NE-1	5.19 ± 0.22	5.10 ± 0.23	5.06 ± 0.27	63.38 ± 3.22	62.50 ± 3.31	60.7 ± 2.20
NE-2	5.49 ± 0.12	5.32 ± 0.16	5.27 ± 0.12	51.04 ± 2.65	50.57 ± 2.99	48.9 ± 3.01
NE-3	5.59 ± 0.14	5.51 ± 0.17	5.45 ± 0.22	35.72 ± 2.70	34.01 ± 2.69	34.19 ± 2.01

**Table 6 cells-10-02404-t006:** Results of pH, viscosity, spreadability, and drug content of glimepiride-loaded (GMP) and inclusion complex-loaded (IC) nanoemulgels (Mean ± SD, *n* = 3).

Parameters	GMP-Loaded Nanoemulgel			GMP/βCD/GEL-44/16-Loaded Nanoemulgel		
	NEG-1	NEG-2	NEG-3	NEG-1	NEG-2	NEG-3
pH	6.20 ± 0.10	6.49 ± 0.09	6.65 ± 0.17	6.16 ± 0.11	6.41 ± 0.16	6.62 ± 0.18
Viscosity (cp)	15,118.71 ± 193.21	15,229.04 ± 137.0	15,375.12 ± 175.6	15,250.9 ± 148.6	15,393.06 ± 177.3	15,475.21 ± 178.2
Spreadability (cm^2^/g)	1.47 ± 0.08	1.43 ± 0.08	1.38 ± 0.05	1.41 ± 0.04	1.37 ± 0.07	1.35 ± 0.08

**Table 7 cells-10-02404-t007:** Permeation parameters of GMP- and GMP/βCD/GEL-44/16-loaded nanoemulsion-based gels (Mean ± SD, *n* = 3).

Formulation Code	Flux (J) (µg/cm^2^/h)	Permeability Constant (Kp) (cm/h)
NEG-1	57.16 ± 7.49 ^abc^	0.028 ^abc^
NEG-2	51.55 ± 3.80 ^abc^	0.025 ^abc^
NEG-3	54.48 ± 6.59 ^abc^	0.027 ^abc^
NEG-1(IC)	70.06 ± 6.60 ^abcdef^	0.035 ^abcdef^
NEG-2(IC)	58.80 ± 3.62 ^abc^	0.029 ^abc^
NEG-3(IC)	62.80 ± 6.66 ^abc^	0.031 ^abc^
GMP in clove oil	10.29 ± 1.25	0.005
GMP/βCD/GEL-44/16 in clove oil	13.86 ± 3.59	0.006
GMP aqueous suspension	3.41 ± 3.25	0.002

^a^ indicates *p* < 0.05 vs. GMP aqueous suspension, ^b^ indicates *p* < 0.05 vs. GMP in clove oil, ^c^ indicates *p* < 0.05 vs. GMP/βCD/GEL-44/16 in clove oil, ^d^ indicates *p* < 0.05 vs. NEG-1, ^e^ indicates *p* < 0.05 vs. NEG-2, ^f^ indicates *p* < 0.05 vs. NEG-3.

## Data Availability

Not applicable.

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
