# Peer review of "Glimepiride-Loaded Nanoemulgel; Development, In Vitro Characterization, Ex Vivo Permeation and In Vivo Antidiabetic Evaluation"

_cells, 2021, doi:10.3390/cells10092404_

Round 1

Reviewer 1 Report

            Conventional delivery of oral antidiabetic drugs faces many problems like poor absorption, low bioavailability, drug degradation, inactivation and the risk of hypoglycemia. Transdermal drug administration is a promising alternative to oral therapy in treatment of diabetes.

            Glimepiride (GMP) is a third-generation sulfonylurea derivative, widely used in the treatment of type 2 diabetes mellitus (T2DM). Transdermal delivery of GMP is hindered due to its low solubility and permeation. Clove oil possesses antidiabetic properties and acts as skin penetration enhancer. Therefore, the authors aimed to formulate nanoemulgel system consisting of GMP or solubility enhanced GMP (GMP-IC) and clove oil as oil phase. The key findings of this study are that (1) synergistic combination of GMP with clove oil improved the skin permeation and hypoglycemic activity of the drug, (2) incorporation of GMP-IC significantly improved release and skin absorption of GMP and its hypoglycemic activity

            Overall, this is a very interesting and well-designed study on an important topic. However, there are some inaccuracies in the manuscript:

  • ABSTRACT:Instead of “non-insulin dependent diabetes mellitus” should be better use – “type 2 diabetes”.
  • INTRODUCTION- There is no spacing before citations [9, 13, 20].
  • MATERIAL AND MATHODS:(a) How many studied groups were involved in the in-vivo study -5 or 6?; (b) How blood was collected for analysis (from the tail vein?); (c) Was glimepiride administered orally by catheter? In order to determine whether the hypoglycemic effect of glimepiride is dependent on stimulation of insulin secretion or on the peripheral potentiation of insulin action, it is advisable to measure insulin levels in addition to glucose levels.
  • RESULTS and DISCUSSION: (a) As the cited publication [12] is not generally available, the abbreviation GMP-IC (GMP-IC= GMP/BCD/GEL-44/16) should be clarified and standardized throughout the text; (b) To facilitate the understanding, the description of Table 8 and Figure 4 should include the full names of the studied groups. An interesting supplement would be to indicate the relationship between different study groups (e.g. 3 and 5; 3 and 6); (c) The markings in the photos are hardly visible (d, H, *).

Reviewer 2 Report

see attached file
